# Information-Theoretic Analysis of Cardio-Respiratory Interactions in Heart Failure Patients: Effects of Arrhythmias and Cardiac Resynchronization Therapy

**DOI:** 10.3390/e25071072

**Published:** 2023-07-17

**Authors:** Mirjana M. Platiša, Nikola N. Radovanović, Riccardo Pernice, Chiara Barà, Siniša U. Pavlović, Luca Faes

**Affiliations:** 1Laboratory for Biosignals, Institute of Biophysics, Faculty of Medicine, University of Belgrade, Višegradska 26-2, 11000 Belgrade, Serbia; 2Pacemaker Center, University Clinical Center of Serbia, University of Belgrade, 11000 Belgrade, Serbia; nikolar86@gmail.com (N.N.R.); pavlosini@yahoo.com (S.U.P.); 3Department of Engineering, University of Palermo, Viale delle Scienze, Building 9, 90128 Palermo, Italy; riccardo.pernice@unipa.it (R.P.); chiara.bara@community.unipa.it (C.B.)

**Keywords:** cardio-respiratory coupling, heart failure, ventricular extrasystoles, CRT responders, information theory, Granger Causality, Transfer Entropy, Cross Entropy

## Abstract

The properties of cardio-respiratory coupling (CRC) are affected by various pathological conditions related to the cardiovascular and/or respiratory systems. In heart failure, one of the most common cardiac pathological conditions, the degree of CRC changes primarily depend on the type of heart-rhythm alterations. In this work, we investigated CRC in heart-failure patients, applying measures from information theory, i.e., Granger Causality (*GC*), Transfer Entropy (*TE*) and Cross Entropy (*CE*), to quantify the directed coupling and causality between cardiac (*RR interval*) and respiratory (*Resp*) time series. Patients were divided into three groups depending on their heart rhythm (sinus rhythm and presence of low/high number of ventricular extrasystoles) and were studied also after cardiac resynchronization therapy (CRT), distinguishing responders and non-responders to the therapy. The information-theoretic analysis of bidirectional cardio-respiratory interactions in HF patients revealed the strong effect of nonlinear components in the *RR* (high number of ventricular extrasystoles) and in the *Resp* time series (respiratory sinus arrhythmia) as well as in their causal interactions. We showed that *GC* as a linear model measure is not sensitive to both nonlinear components and only model free measures as *TE* and *CE* may quantify them. CRT responders mainly exhibit unchanged asymmetry in the *TE* values, with statistically significant dominance of the information flow from *Resp* to *RR* over the opposite flow from *RR* to *Resp*, before and after CRT. In non-responders this asymmetry was statistically significant only after CRT. Our results indicate that the success of CRT is related to corresponding information transfer between the cardiac and respiratory signal quantified at baseline measurements, which could contribute to a better selection of patients for this type of therapy.

## 1. Introduction

The coordination between beat-to-beat (*RR*) interval and respiratory signals is crucial for maintaining homeostasis in the coupled cardio-respiratory systems. In the last several years, approaches proposing unidirectional and bidirectional quantities of relationship between cardiac cycles and respiratory rhythm have been increasingly used, with potential applicability in revealing new insights in many physiological and pathological conditions [1,2,3,4,5,6,7,8,9,10].

Many of these approaches work in the so-called framework of information dynamics, which provides measures able to assess the ‘information content’ of individual dynamic processes or collections of processes, and the information exchange among them [9,11,12,13]. A dynamical approach takes into account the flow of time by investigating how much the past system history contributes to reduce the uncertainty about the present state [11]. In this context, well-established tools like Granger Causality (*GC*) and Transfer Entropy (*TE*) assess pairwise directional interactions between two subsystems of a complex system, quantifying the directed flow of information from one subsystem to another [14,15,16,17]. On the other hand, measures of cross-predictability like cross-entropy (*CE*) quantify how much one process can be predicted from the other, returning asymmetric measures of coupling without implementing the Granger concept of causality [18]. All these measures have been widely applied in physiological contexts, e.g., for investigating cardiovascular variability [19,20], but also for studying cardiorespiratory interactions in healthy subjects [11] or in patients suffering of various types of sleep apnea [9].

In previous works, we investigated cardio-respiratory interactions in heart-failure patients using unidirectional markers and bidirectional markers based on linear models at baseline measurements [6,21]. The potential of the information-theoretic approaches described above has still not been tested for evaluating the effects of cardiac resynchronization therapy (CRT) in heart failure (HF). CRT has been used for more than 20 years in the treatment of symptomatic patients with heart failure with reduced left ventricular ejection fraction (LVEF) and abnormal QRS duration and morphology [22,23]. CRT leads to restoration of electromechanical synchrony, resulting in an improvement in the cardiac pump function, ejection fraction and reverse remodeling of the left ventricle. In most patients, these mechanical effects are associated with an improvement in the functional capacity and with a reduction in morbidity and mortality [24]. Nowadays, we know that, in addition to restoring ventricular mechanical synchronization, there are various cardiac and extra-cardiac effects of CRT, which are also responsible for its beneficial outcomes. One such effect is the remodeling of β-adrenergic signaling pathways and the restoration of sympathovagal balance [25]. The effect of CRT on cholinergic signaling has not been fully investigated in the literature yet, nor has the significance of CRT-induced neurohumoral modulation and its influence on cardio-respiratory interactions in these patients [26]. In recent years, significant advancements have been made in patient selection, implantation approaches, implant material performances, post-implantation device programming and medical therapy optimization. However, still a third of patients do not achieve the expected benefit from CRT implantation [27]. Therefore, it is very important—but also challenging—to use new approaches in the analysis of heart-failure patients who are candidates for CRT implantation, to preoperatively separate future responders from non-responders, according to parameters that have not yet been used. In this context, parameters that define autonomic function and cardio-respiratory interactions can certainly be of great importance.

In this work, we evaluate the cardio-respiratory interactions in terms of information flows between the coupled *RR* and respiratory signal dynamics. We used both the linear parametric and the nonlinear model-free implementations of the concept of Granger causality provided by *GC* and *TE* measures, as well as cross entropy, to investigate asymmetric coupling on different HF CRT responders and non-responders.

## 2. Materials and Methods

### 2.1. Subjects and Experimental Protocol

The data analyzed in this study consist in part of the raw data obtained from previously recruited patients with heart failure (reduced left ventricular ejection fraction, LVEF < 35%), and indication for CRT device implantation published in our previous work where the pool of results from all patients were analyzed together [21]. The study was approved by the Ethics Committee of the Faculty of Medicine the University of Belgrade, and each subject signed an informed consent form (Approve Date: 17 March 2017, Ref. Numb.29/III-4). Since the techniques applied in this work are sensitive to the type of heart rhythm, we excluded data from 8 patients (2 responders to CRT) with permanent atrial fibrillation from the group of 47 heart-failure patients. The main reason is that, since the cardiac rhythm in atrial fibrillation is highly irregular, the corresponding data would not satisfy stationarity criteria needed for our analysis; moreover, we have previously shown that the influence of respiration on cardiac rhythm and in the opposite direction is very small in patients with atrial fibrillation [6]. Contrary to our previous study [21] where we did not separate groups by arrhythmias, in this study we examined the effects of a low and high number of VESs on cardio-respiratory coupling before and after CRT. For this study, we divided the analyzed 39 HF patients into 3 groups: a first group of 14 patients with sinus rhythm (HFSin), a second group of 11 patients with a low number (<6) of ventricular extrasystoles (HFVES1), and a third group of 14 patients with 6 and more ventricular extrasystoles (HFVES2). Measurements were performed before (baseline) and approximately 9 months after CRT device implantation (follow-up). After follow-up, patients were divided into two groups, i.e., responders (*N* = 25) and non-responders (*N* = 14), in relation to the response to CRT, which was assessed according to changes in certain clinical and echocardiographic parameters. We defined an echocardiographic responder to CRT as a patient with increased left ventricular EF by 5% or more, and functional (clinical) who improved at least one class category in the New York Heart Association functional classification and/or the six-minute walk test by at least 10% at follow-up. In this study, a responder to resynchronization therapy had to meet both echocardiographic and functional criteria. Both baseline and follow-up experiments were conducted in the morning, between 7 and 8 a.m., in a quiet room surrounding at the Pacemaker Center of the University Clinical Center of Serbia. Baseline measurements were carried out immediately before device implantation. Data were acquired from 20 min of electrocardiographic (ECG) and respiratory (*Resp*)-signal measurements of relaxed subjects in the supine position and at a spontaneous breathing frequency by the Biopac MP100 system and AcqKnowledge 3.9.1 software (BIOPAC System, Inc., Santa Barbara, CA, USA) using a sampling rate equal to 1 kHz [21]. ECG data were acquired using the ECG 100C electrocardiogram amplifier module, while an RSP 100C respiratory pneumogram amplifier module with TSD 201 transducer attached to the belt (using an adjustable nylon strap) was used to measure abdominal expansion and contraction. The interbeat interval (i.e., ECG *RR* intervals) time series in patients with sinus rhythm and sinus rhythm with ventricular extrasystoles were analyzed. Interbeat (*RR*) intervals and interbreath (*BB*) intervals were extracted from signals recorded using the tool Pick Peaks from OriginPro 8.6 (OriginLab Corporation, Northampton, MA, USA). In the ECG with ventricular extrasystoles (VES), we used the time coordinate of the peak from VES as R peak coordinate for the subsequent analyses.

The first analysis consisted in evaluating the changes in *RR* and *BB* intervals by computing their mean and standard error. Furthermore, given that the *RR* and respiration samples obtained in this way were unequally positioned, an equal equidistant resampling of both series was carried out using the mean *RR* value of each individual as the resampling interval. The resampling procedure was performed via linear interpolation between two corresponding adjacent existing samples [3]. In this way, the resampling frequency was different for each subject, and fell within the range (0.8–1.5) Hz. The final analyzed *RR* and *Resp* time series were measured from signal lasting 20 min, resulting in a different numbers of samples according to the individually varying resampling frequency (Table 1).

### 2.2. Information-Theoretic Measures and Data Analysis

For the analysis of *RR* and *BB* intervals, we computed the mean and standard error taking into account HFSin, HFVES1, HFVES2 groups, both at baseline and at follow-up for both responders and non-responders to CRT. Then, to investigate the dynamics of coupling and causality within the cardio-respiratory system, the information-theoretic measures of *GC*, *TE* and *CE* were computed on the *RR* and *Resp* time series. 

These measures are defined, in the context of dynamic systems mapped by random processes, considering a bivariate random process $S=\left\{X,Y\right\}$ composed of two interacting processes *X* and *Y*, respectively, considered as the driver and target process. In this context, the concept of Granger causality from *X* to *Y* is formalized through the measure of Transfer Entropy, which quantifies the information that the past states of driver process, $X_{n}^{-}=[X_{n-1},X_{n-2},\ldots ]$, transfer to the present state of the target process, $Y_{n}$, when the past states of the target itself, $Y_{n}^{-}=[Y_{n-1},Y_{n-2},\ldots ]$, are known. The *TE* is defined as:(1)$$TE_{X\to Y}=I\left(Y_{n};\left.X_{n}^{-}\right|Y_{n}^{-}\right)$$
where $I\left(Y_{n};\left.X_{n}^{-}\right|Y_{n}^{-}\right)$ is the conditional mutual information. On the other hand, the predictability of the present state of the target process $Y_{n}$ from the past states of the driver process $X_{n}^{-}$ is quantified through the cross entropy defined as: (2)$$CE_{X\to Y}=I\left(Y_{n};X_{n}^{-}\right)$$
where $I\left(Y_{n};X_{n}^{-}\right)$ is the mutual information.

In addition to the standard definitions given in (1) and (2), when assessing influences within physiological systems, it may be important to take into account the so-called instantaneous effects, i.e., the influence that the current state of driver process $X_{n}$ can have on the target state $Y_{n}$ [28,29]. To this end, the definitions of *TE* and *CE* can be modified as follows [28]:(3)$$iTE_{X\to Y}=I\left(Y_{n};X_{n},\left.X_{n}^{-}\right|Y_{n}^{-}\right)$$
(4)$$iCE_{X\to Y}=I\left(Y_{n};{X_{n},X}_{n}^{-}\right)$$

The so-called instantaneous *TE* and instantaneous *CE* defined in (3) and (4) are used when the instantaneous effects are deemed as physiologically relevant, i.e., in the direction from *Resp* to *RR* but not in the direction from *RR* to *Resp*.

In this work, the above presented measures were implemented in practice using both a linear model-based and a non-linear model-free estimator. The first approach makes use of linear regression models whereby the present state of the target process $Y_{n}$ is described as a linear combination of its *p* past states, $Y_{n}^{p}=[Y_{n-1},\ldots ,Y_{n-p}]$, or of the *p* past states of both processes, $\left[X_{n}^{p},Y_{n}^{p}\right]$. These linear regression models return prediction errors, or residuals, whose variance can be related to the concept of conditional entropy under the assumption of Gaussianity and can be exploited to provide a measure of predictability improvement related to the *TE* [11]. Specifically, denoted as $\sigma_{\left.Y_{n}\right|Y_{n}^{p}}^{2}$, and $\sigma_{\left.Y_{n}\right|X_{n}^{p},Y_{n}^{p}}^{2}$ the prediction error variances of the two regression models, the measure of *GC* was computed as [30]
(5)$$GC_{X\to Y}=\ln\frac{\sigma_{\left.Y_{n}\right|Y_{n}^{p}}^{2}}{\sigma_{\left.Y_{n}\right|X_{n}^{p},Y_{n}^{p}}^{2}}$$

Moreover, an extended measure of *GC* incorporating the instantaneous effect from X to Y and denoted as $iGC_{X\to Y}$ was computed, augmenting the vector $X_{n}^{p}$ with the inclusion of $X_{n}$ in (5) [31].

The second method is a model-free approach that makes use of nearest-neighbors to compute the probability density functions needed for the estimation of entropy measures [32]. Here, we used the formulation reported in [33] which performs, for each pattern forming a realization of the present and past states of the analyzed processes, the search for its *k* nearest neighbors to compute the entropy in the highest-dimensional space, and then adopts a distance-projection strategy to compute the entropies in the spaces of lower dimension. The resulting estimator has the form (see [34] for details)
(6)$$TE_{X\to Y}=\psi \left(k\right)+\left\langle \psi \left(N_{Y_{n}^{q}}+1\right)-\psi \left(N_{Y_{n}Y_{n}^{q}}+1\right)-\psi \left(N_{X_{n}^{q}Y_{n}^{q}}+1\right)\right\rangle$$
where $\psi \left(\cdot \right)$ is the digamma function, k is the number of neighbors considered in the highest dimensional space, $\varepsilon_{n,k}$ is twice the distance of the *n*-th reference pattern $\left(y_{n},y_{n}^{q},x_{n}^{q}\right)$ to its kth neighbor in the highest dimensional space, and $N_{X_{n}^{q}Y_{n}^{q}}$, $N_{Y_{n}Y_{n}^{q}}$ and $N_{Y_{n}^{q}}$ are the number of patterns whose distance from the projected reference pattern ($(y_{n}^{q},x_{n}^{q}$), ($y_{n},y_{n}^{q}$) and $y_{n}^{q}$, respectively) is lower than $\varepsilon_{n,k}/2$. A similar treatment leads to estimating the cross entropy measure as:(7)$$CE_{X\to Y}=\psi \left(k\right)+\psi \left(N\right)-\left\langle \psi (N_{Y_{n}}+1)+\psi \left(N_{X_{n}^{q}}+1\right)\right\rangle$$
where N is the number of available patterns and the other symbols have the same meaning as in (6) (thus $N_{X_{n}^{q}}$ and $N_{Y_{n}}$ are the number of patterns whose distance from the projected $x_{n}^{q}$ and $y_{n}$ is lower than $\varepsilon_{n,k}/2$). As for the linear estimation of *GC*, extended measures of *TE* and *CE* were also obtained augmenting the vector $X_{n}^{q}$ with the inclusion of $X_{n}$ in (6) and (7) when the instantaneous effect from X to Y was deemed as causally relevant.

In the analyzed dataset, the measures defined above were computed along the two directions of interaction from *Resp* to *RR* and from *RR* to *Resp*, to analyze closed-loop bidirectional cardiorespiratory interactions. Given the adopted measurement convention, the heartbeat cannot transfer information at zero-lag to the respiratory system, since the *i*-th breath sample is simultaneous with the onset of the *i*-th cardiac period. Therefore, the standard measures of *GC*, *TE* and *CE* given in (5)–(7) computed without incorporating the instantaneous effect were used when *X* = *RR* and *Y* = *Resp*, while the extended measures in which the vector of the past driver states is augmented with the present driver state were used when *X* = *Resp* and *Y* = *RR*.

With regard to *GC* computation, the model order *p* was set according to the Akaike information criterion (AIC), taking care to consider the present state of the driver in defining the model in the direction of analysis from *Resp* to *RR*. Even if the choice of the model order can influence the value of the achieved information measures and thus their physiological significance, in this work a sufficiently high value was selected in order to have enough ‘memory’ of the two processes involved in the interaction [34].

Regarding the non-linear implementation of *CE* and *TE* measures, the number of neighbors *k* was fixed to 10 and the number of considered past states *q* to 2, as commonly performed in the literature in the field [35].

### 2.3. Statistical Analysis

In order to assess the statistical significance of the differences of the various indexes between groups and conditions, we employed nonparametric tests. For each measure, the distributions of the results were tested against normality through a Shapiro–Wilk test and since the most of the measures did not follow a normal distribution, we performed nonparametric tests for their comparison. 

We performed the Wilcoxon test to compare the mean values of sample numbers in each group before and after CRT, and the Mann–Whitney U test to compare CRT groups in each condition. The Kruskal–Wallis H test was applied to compare the mean values among the HF groups (HFSin, HFVES1 and HFVES2) in each condition (Table 1).

For *RR* and *BB* intervals analyses we tested (1) the baseline versus follow-up condition using the Wilcoxon test for each HF group, and the CRT group; (2) responders versus non-responders performing the Mann–Whitney U test in both conditions; and (3) differences between the HF groups, before and after CRT, using also the Mann–Whitney U test (*n* = 3: HFSin vs. HFVES1, HFSin vs. HFVES2, and HFVES1 vs. HFVES2), Table 2.

For information-theoretic measures, the following statistical analyses were performed in HF groups and in CRT groups separately. We used the Mann–Whitney U test to compare groups at baseline and at follow-up measurements. In each group, significant differences between bidirectional measures distributions were recognized by a Wilcoxon non-parametric pairwise test. For comparison of these parameters between baseline and follow-up measurements, we also used a Wilcoxon test.

For all analyses, probability values *p* < 0.05 were considered statistically significant. Statistical analyses were carried out in SPSS (SPSS Inc., Chicago, IL, USA) version 17.

## 3. Results

In Table 2, the analysis of *RR* intervals and *BB* intervals in the analyzed groups is shown, with regard to descriptive statistics measures (mean and standard error). Responders to CRT exhibited statistically significantly prolonged *RR* intervals (*p* = 0.028) and *BB* intervals (*p* = 0.014) compared to baseline, while non-responders did not change these markers (*p* > 0.05). Also, there was no difference between the responder and non-responder groups, except for *BB* intervals at follow-up measurements (*p* = 0.013).

Figure 1, Figure 2 and Figure 3 present, as bar graphs, the results obtained with regard to linear and nonlinear measures of *GC*, *TE*, and *CE* computed along the two directions of interaction from *Resp* to *RR* and from *RR* to *Resp*. There was a tendency for *GC*(*Resp-RR*) to be higher than *GC*(*RR-Resp*) in all patients at baseline condition, but only after CRT device implantation in the HFSin and HFVES2 group the difference was statistically significant, with *p* < 0.05 (Figure 1). 

Moreover, compared with the baseline measurement, in the group of HFSin patients, CRT showed significantly decreased *GC*(*RR-Resp*), *p* = 0.048 (not shown in Figure 1), resulting in a significant difference between *GC*(*Resp-RR*) and *GC*(*RR-Resp*) (*p* < 0.05) (Figure 1B). In the group of HFVES1 patients, there was no statistically significant difference between *GC* in both measurements (Figure 1).

In Table 3, the results of the comparisons between coupling and causality measures computed in the HF groups at baseline and at follow-up measurements are presented. Linear Granger causality measures were sensitive to a higher number of VESs and at baseline measurements they significantly decreased with the number of VESs in both directions. After CRT implementation, a statistically significant difference was found in *GC*(*Resp-RR*) between the HFSin patients’ group and the two other groups and in *GC*(*RR*-*Resp*) between the HFSin and the HFVES2 group.

Figure 2 shows the measures of *TE* computed along the two directions of interaction between *Resp* and *RR*. A statistically significant difference between *TE*(*Resp-RR)* and *TE*(*RR-Resp*) in the HFSin and HFVES2 groups of patients was present at the baseline and was maintained after CRT implantation (*p* < 0.01), also showing a tendency to become more marked. Only in the HFVES1 group, *TE*(*Resp-RR*) was not significantly greater than *TE*(*RR-Resp*). In the comparison between groups at the baseline measurement, we found a statistically significant difference in *TE*(*RR-Resp*) between groups of patients with VES (Table 3). CRT influenced *TE*(*RR-Resp*) in the HFSin and HFVES1 group with decreasing values but only towards to the tendency of statistical significance (*p* = 0.08) and (*p* = 0.12).

Contrary to *GC* and *TE*, where *RR* to *Resp* measures were lower than *Resp* to *RR* measures, *CE*(*RR-Resp*) were higher than *CE*(*Resp-RR*) in all analyzed groups (Figure 3). Statistically significant differences were preserved only in HFSin group during both baseline and follow-up. In the other two groups, statistically significant differences were reported only after CRT implantation. Comparison between groups did not reveal any statistically significant difference in *CE* measures.

Figure 4 presents the results of the comparison between baseline and follow-up markers and between the two directions, in both groups of responders and non-responders. In the group of CRT responders, the two *GC* measures did not change after CRT, while in the group of non-responders, follow-up measurements indicated a reduction in *GC*(*RR-Resp*). In this group, although unchanged, *GC*(*Resp-RR*) became significantly higher than *GC*(*RR-Resp*) after the follow-up measurement. Further, *TE*(*Resp-RR*) was significantly higher than *TE*(*RR-Resp*) in both baseline and follow-up measurements in responders, and in non-responders only after follow-up measurement. Contrary, *CE*(*Resp-RR*) was significantly lower than *CE*(*RR-Resp*) in both measurements in responders and in non-responders only after follow-up measurement. No significant differences between directed measures for any measure were found at baseline in non-responders while significant differences were obtained for all information-theoretic measures at baseline in the group of CRT responders.

## 4. Discussion

### 4.1. Heart Rate and Respiratory Rate

The analysis of the mean heart and respiratory rates confirms the results from previous works, and here it is complemented with a dynamic analysis of the variability of *RR* and *BB* intervals [6]. The results of our research indicate that in patients with heart failure, resting heart and respiratory rates are the highest in patients with a large number of ventricular extrasystoles before CRT. When we separately analyze CRT responders and non-responders, we observe that during follow-up there is a decrease in heart rate in all HF patients, but that it is statistically significant only in CRT responders. The reasons for this result are related to the intensification of antiarrhythmic therapy in all patients after CRT implantation, as well as to the improvement in clinical status and the recovery of vagal tone, which dominantly determines this parameter, in patients who benefited from device implantation [36,37].

Respiratory rate also significantly decreases in CRT responders during follow-up, but increases slightly in non-responders, resulting in a statistically significant difference in the values of this parameter between these two groups at control examination. An improvement in cardiac function enables better perfusion, i.e., a reduction in tissue hypoxia, which leads to reduction in chemoreceptor stimulation, and this can explain the decrease in respiratory rate in CRT responders [38]. Our results also confirm that better functional capacity, according to the NYHA (New York Heart Association) classification of HF, and higher LVEF, are associated with lower respiratory frequency [39].

### 4.2. Granger Causality and Transfer Entropy

In this study the model-based Granger causality analysis showed a stronger influence of respiration on heart rhythm in all HF patients. The reasons for small discrepancies in *GC* analysis comparing these results and the findings reported in our previous study in the HF groups are probably methodological, since we used a different *GC* algorithm in the last paper [40]. In addition, the groups with VES were not equally defined: in our previous work, the HFVES group was defined by more than 20 premature ventricular ectopic beats during signal recording [6]. According to Granger causal analyses, such as the *GC* measure (model-based) and the *TE* measure (model-free), it is expected that respiration would have a stronger impact on the heart rate than vice versa. This result can be documented in healthy subjects and describes the well-known phenomenon of respiratory sinus arrhythmia (RSA) whereby *Resp* changes modulate the heart period duration differently during inspiration than during expiration. Our results, for the HF patients in sinus rhythm, document this effect only when the *TE* was used (Figure 2A), but not using the *GC* (Figure 1A), suggesting that RSA mechanisms are nonlinear and cannot be fully captured using the linear *GC* analysis.

Furthermore, the *GC* decreased progressively from sinus rhythm to the HFVES1 and the HFVES2 group during baseline (Figure 1A), while the *TE* did not show an evident similar trend (Figure 2A). This suggests that in HF patients, the cardiorespiratory interactions evaluated in the presence of extrasystoles have a strong nonlinear component which can be properly captured only when *TE* is estimated.

A similar influence of cardiac rhythm on respiration and vice versa in the HFVES1 group at the control recording is difficult to interpret. The effect of respiration on cardiac rhythm is determined by various factors even under physiological conditions. For instance, it has been shown that this influence loses its importance in the elderly, i.e., the strength of direct respiratory modulation of cardiac rhythm decreases with aging, as well as the indirect effects of respiration on heart rate due to the reduction in baroreflex sensitivity [41,42,43,44]. When autonomic imbalance occurs (as in HF), this bidirectional interaction is certainly altered, with the activation of compensatory mechanisms that are still poorly understood. In the previous paper [6], we presented the assumption that the stronger influence of the heart rhythm on the respiratory signal in patients with HF and VES is due to compensatory mechanisms that restore the regularity of the heart rhythm and increase its influence on breathing. However, after this study, it is clear that the frequency of extrasystoles, as well as possible other factors such as the HF stage or age, have an impact on bidirectional cardio-respiratory interactions. On the other hand, we observed that during follow-up, the influence of breathing on heart rhythm significantly weakened in patients with a low number of VESs, thus demonstrating a stronger influence of the cardiac rhythm on the respiratory signal in the HFVES1 group.

Regarding the *TE* values in HFVES patients, we noticed that causality along the direction from the *RR* to *Resp* in the follow up measurements was not sensitive to the higher frequency of extrasystoles, as it was in baseline recordings (Table 3). On the other hand, the prevalence of the causal interaction along the direction *Resp* to *RR* becomes much more marked after CRT, and this was detected using both *GC* and *TE* (Figure 1B and Figure 2B). This suggests that resynchronization therapy is successful in restoring RSA in HF patients, activating both the linear and nonlinear components of the interaction from *Resp* to *RR*. The recovery of vagal tone and baroreceptor activity is responsible for strengthening the effect of respiration on cardiac rhythm, i.e., that intense cardiorespiratory interaction in this direction is a confirmation of the existence of the balanced activity of control and compensatory mechanisms. This result also implies that with the restoration of *Resp* on *RR,* we can detect the joined mechanism of the reduced linear component of cardiac causality from *RR* on restored *Resp* (Figure 1A,B).

### 4.3. Cross Entropy

The opposite trends exhibited by *CE* if compared to *GC* and *TE* reflect a prevalence of the coupling in the direction *RR* to *Resp*, amplified after CRT (Figure 3). The reason for this difference is very likely methodological: while *GC* and *TE* quantify directed interactions in terms of predictability improvement, the *CE* implements the concept of cross-predictability. From this point of view, the *CE* is more similar to measures like convergent cross-mapping, which attempt to infer the dominant causation in a bidirectional interaction by predicting the driver starting from the history of the target process [45]. Therefore, according to this methodological consideration, it is not surprising that a predominant *Resp* to *RR* mechanism produces a higher *GC*/*TE* from *Resp* to *RR* but also higher *CE* from *RR* to *Resp*.

The values of *CE*(*Resp-RR*) and *CE*(*RR-Resp*) were not significantly different between the groups at any of the measurements, and both parameters had the largest preoperative values in the HFVES1 group, and at control recording in the HFSin group. This suggests that in cardiorespiratory coupling, the prediction of the dynamics of one signal based solely on the history of the other is stable in time and that the presence of arrhythmias does not significantly affect it.

Our finding that in HF patients, cardio-respiratory interactions assessed in the presence of ventricular extrasystoles have a strong nonlinear component that can only be properly captured when using *TE,* was confirmed by the results of model-free *CE*, which did not exhibit any particular trends, and after the resynchronization therapy clearly differentiated the two directions of interaction even in presence of extrasystoles. Transfer entropy, in addition to confirming that the occurrence of ventricular extrasystoles significantly affects the interactions of the respiratory and cardiovascular system, also indicates that their frequency decisively determines the intensity of information flow between these systems. What compensatory mechanisms and at what levels of regulation of physiological processes trigger the occurrence of ventricular extrasystoles, and what are the causes of their pronounced dynamics during monitoring, as well as in relation to the extra beats frequency, remain to be determined by future research.

### 4.4. Cardiac Resynchronization Therapy

The results of the comparative analysis of CRT responders and non-responders provide useful insights about the reaction of patients to therapy. At the baseline measurement, no significant differences between these two groups were reported with regard to Granger causality analysis, with a stronger influence of respiration on heart rhythm in both groups of patients at the control examination in the case of the model-free approach (Figure 4C,D) and only for non-responders with the linear measure (Figure 4B). 

Moreover, it is noteworthy that the nonlinear measure of causality is able to identify the prevalence of causality from respiration to cardiac activity in the baseline condition for responders to CRT (Figure 4C). This finding may suggest that if there is a statistically significant difference between *TE* parameters in a patient who is a candidate for CRT, he/she will benefit from device implantation. However, this would be too bold and probably a wrong conclusion, given that in CRT non-responders, the difference between these parameters preoperatively is close to statistical significance, and is achieved at the follow-up examination.

The values of both CE parameters were lower in CRT responders compared to non-responders for all the measurements. During follow-up, statistically significantly higher values of *CE*(*RR*-*Resp*) compared to *CE*(*Resp*-*RR*) were observed in CRT responders and non-responders.

It is also very interesting to observe changes in the intensity of the information flow in different directions in CRT non-responders. In fact, the *TE*(*Resp-RR*) increase in patients with significantly reduced vagal tone is an indicator that the values of RSA, which is a measure of cardiorespiratory interaction, and *CE*(*Resp-RR*), which is a measure of cardio-respiratory coupling, do not necessarily follow each other, and this is emphasized in some previous works [46].

## 5. Conclusions

Information-theoretic analysis of bidirectional cardio-respiratory interactions revealed the strong effect of nonlinear components in the *RR* and *Resp* time series, as well as in their causal interaction in evaluation of cardiac resynchronization therapy in HF patients. 

Our results, for the HF patients in sinus rhythm, document the RSA effect only when using the *TE* but not *GC*. On the other hand, the prevalence of the causal interaction along the direction from *Resp* to *RR* becomes much more marked after CRT, and this was detected using both *GC* and *TE,* which suggests that the resynchronization therapy is successful in restoring RSA in HFSin patients, activating both the linear and nonlinear components of the interaction from *Resp* to *RR*. Regarding the *CE*, it exhibits opposite trends of *GC* and *TE*, showing prevalence of coupling in the direction from *RR* to *Resp*, enhanced after CRT. 

In the HF patients, before and after CRT, the occurrence of a high number of extrasystoles significantly reduced *GC* in both directions. The *GC* progressively decreases from HF in sinus rhythm to HFVES1 and to HFVES2, while the *TE* does not show a similar evident trend. This finding suggests that in HF patients, the cardio-respiratory interactions evaluated in the presence of extrasystoles have a strong nonlinear component that can only be properly captured when the *TE* is used. This is confirmed also by the *CE* formulated in a model-free way, which does not show particular trends and after the resynchronization therapy clearly differentiates the two directions of interaction even in the presence of extrasystoles.

Contrary to results obtained in the group of non-responders, in the group of responders to CRT we found a statistically significant difference in the baseline condition between the *Resp*-*RR* and *RR*-*Resp* directions, quantified by applied measures from information theory. This finding could be useful in the selection of patients for CRT.

## Figures and Tables

**Figure 1 entropy-25-01072-f001:**
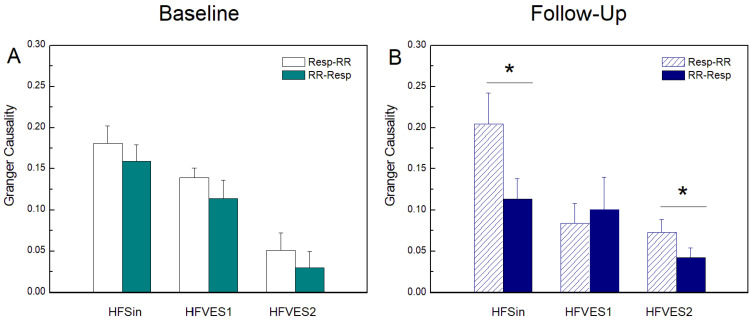
Granger causality (*GC*) at baseline measurements (**A**) and at follow-up measurements after CRT device implantation (**B**) in heart-failure patients: with sinus rhythm (HFSin), with sinus rhythm and small number of ventricular extrasystoles (HFVES1), and with sinus rhythm and higher number of ventricular extrasystoles (HFVES2). *GC*(*Resp-RR*) and *GC*(*RR-Resp*) denote causality of respiration in *RR* intervals time series and vice versa. Data are presented as mean + standard error. * *p* < 0.05.

**Figure 2 entropy-25-01072-f002:**
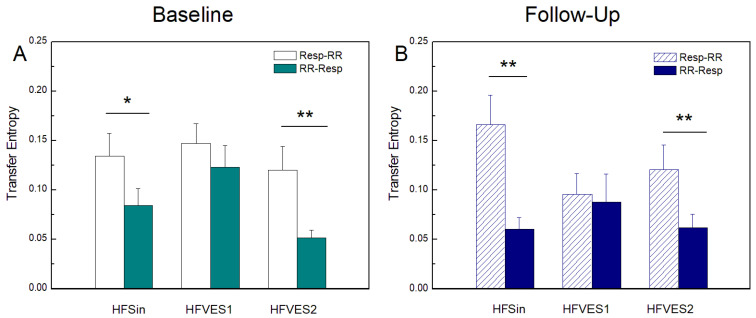
Transfer entropy (*TE*) at baseline measurements (**A**) and at follow-up measurements after CRT device implantation (**B**) in heart-failure patients: with sinus rhythm (HFSin), with sinus rhythm and small number of ventricular extrasystoles (HFVES1), and with sinus rhythm and higher number of ventricular extrasystoles (HFVES2). *TE*(*Resp-RR*) and *GC*(*RR-Resp*) denote causality of respiration in RR intervals time series and vice versa. Data are presented as mean + standard error. ** *p* < 0.01, * *p* < 0.05.

**Figure 3 entropy-25-01072-f003:**
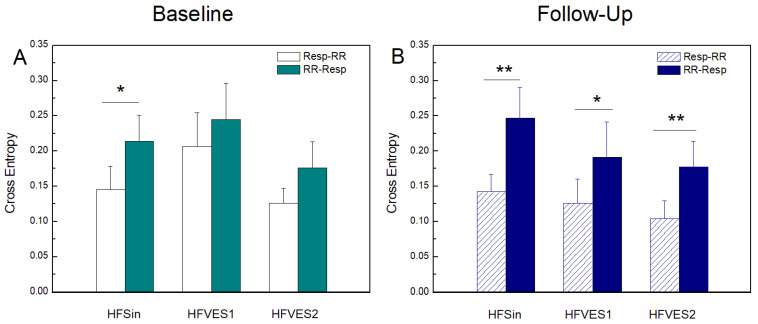
Cross entropy (*CE*) at baseline measurements (**A**) and at follow-up measurements after CRT device implantation (**B**) in heart failure patients: with sinus rhythm (HFSin), with sinus rhythm and small number of ventricular extrasystoles (HFVES1), and with sinus rhythm and higher number of ventricular extrasystoles (HFVES2). *TE*(*Resp-RR*) and *CE*(*RR-Resp*) denote causality of respiration in RR intervals time series and vice versa. Data are presented as mean + standard error. ** *p* < 0.01, * *p* < 0.05.

**Figure 4 entropy-25-01072-f004:**
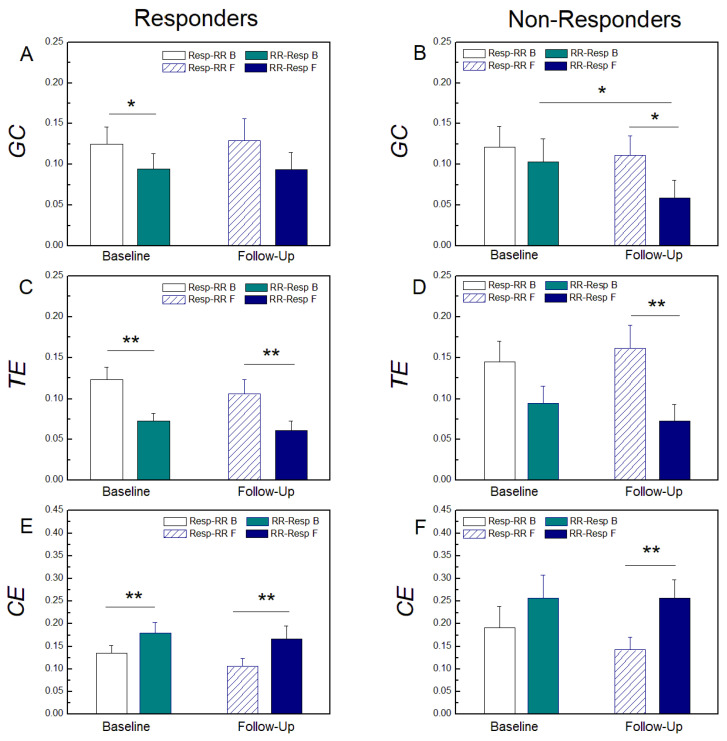
Granger causality (*GC*) at baseline measurements and at follow-up measurements in responders (**A**) and non-responders (**B**) to CRT. Transfer Entropy (*TE*) at the baseline measurements and at the follow-up measurements in responders (**C**) and non-responders (**D**) to CRT. Cross Entropy (*CE*) at the baseline measurements and at the follow-up measurements in responders (**E**) and non-responders (**F**) to CRT. *Resp-RR* and *RR-Resp* denote causality of respiration in RR interval time series and vice versa. Data are presented as mean + standard error. ** *p* < 0.01, * *p* < 0.05.

**Table 1 entropy-25-01072-t001:** Comparison of the mean values of sample numbers in each group before (baseline) and after (follow-up) cardio resynchronization therapy (CRT) as well as of the mean values between the groups in each condition.

Group	Baseline (×10^3^)	Follow-Up (×10^3^)	*p* (Baseline vs. F-Up)
HFSin (*N* = 14; 71% R)	1.31 ± 0.18	1.22 ± 0.19	0.124
HFVES1 (*N* = 11; 64% R)	1.34 ± 0.29	1.25 ± 0.22	0.131
HFVES2 (*N* = 14; 57% R)	1.49 ± 0.18	1.30 ± 0.17	0.008
*p* (Among groups)	0.061	0.469	
Responders (*N* = 25)	1.43 ± 0.22	1.28 ± 0.18	0.007
Non-Responders (*N* = 14)	1.30 ± 0.22	1.22 ± 0.21	0.090
*p* (Resp. vs. Non-Resp.)	0.118	0.149	

Values are mean ± standard deviation. In the brackets percent of responders (R) in each HF group is given.

**Table 2 entropy-25-01072-t002:** Descriptive statistics.

			HF Groups		CRT Groups	
	Condition	HFSin	HFVES1	HFVES2	Responders	Non-Responders
*RR* [s]	Baseline	0.930 ± 0.032 ^ττ^	0.932 ± 0.054	0.813 ± 0.026 **	0.857 ± 0.027 *	0.943 ± 0.040
	Follow-up	1.010 ± 0.053	0.99± 0.40	0.934 ± 0.031	0.959 ± 0.036	1.007 ± 0.051
*BB* [s]	Baseline	4.25 ± 0.38	3.82 ± 0.27	3.40 ± 0.13	4.01 ± 0.24 *	3.49 ± 0.14
	Follow-up	4.30 ± 0.39	3.96 ± 0.37	3.65 ± 0.21	4.32 ± 0.26 #	3.35 ± 0.14

** *p* < 0.01 * *p* < 0.05 Baseline vs. Follow-up, # *p* < 0.05 Responders vs. Non-Responders, ^ττ^
*p* < 0.01 HFSin vs. HFVES2.

**Table 3 entropy-25-01072-t003:** Statistical significance of comparisons between pairs of groups for bidirectional measures.

		Baseline			Follow-Up	
	HFSin vs. HFVES1	HFSin vs. HFVES2	HFVES1 vs. HFVES2	HFSin vs. HFVES1	HFSin vs. HFVES2	HFVES1 vs. HFVES2
*GC*(*Resp-RR*)	0.432	**0.001**	**0.002**	**0.013**	**0.004**	0.751
*GC*(*RR-Resp*)	0.297	**0.001**	**0.001**	0.212	**0.012**	0.527
*TE*(*Resp-RR*)	0.572	0.635	0.165	0.080	0.246	0.681
*TE*(*RR-Resp*)	0.181	0.137	**0.005**	1.000	0.946	0.918
*CE*(*Resp-RR*)	0.258	0.131	0.123	0.440	0.781	0.758
*CE*(*RR-Resp*)	0.643	0.274	0.258	0.411	0.980	0.957

Statistically significant values are in bold.

## Data Availability

Data will be made available on request.

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
