# Peer review of "Information-Theoretic Analysis of Cardio-Respiratory Interactions in Heart Failure Patients: Effects of Arrhythmias and Cardiac Resynchronization Therapy"

_entropy, 2023, doi:10.3390/e25071072_

Round 1

Reviewer 1 Report

The paper “Information-theoretic analysis of cardiorespiratory interactions in heart failure patients: effects of arrhythmias and cardiac re-synchronization therapy” describes the estimation of the cardiorespiratory coupling by linear and nonlinear approaches in patients suffering heart failure. The goals were to determine the effects of cardiac resynchronization therapy (CRT) on cardiorespiratory interaction and if this interaction could help to differentiate responders and non-responders to CRT in a preoperative phase. For that purpose, the ECG and the respiratory signal were measured immediately before CRT and after nine months. The paper is well-written and organized, representing a follow-up of a previous effort by the authors.  Some questions and comments are as follows:

·   1) According to reference 21, the CRT responders and non-responders were 62.9% and 80% men, respectively. Although there is no statistical difference in sex, according to Table 1 of the paper, the prevalence of men in the non-responders calls attention. Also, there is a statistical difference (p<0.02) for ischemic heart disease, with 50% in the non-responders group. What could the authors add about the influence of these two issues in the analyzed cardiorespiratory coupling?

·  2) Regarding acquiring the respiratory signal using a belt to measure abdominal movements, according to the literature, women have thoracoabdominal respiration, i.e., women and men have differences in respiratory movements. How does this impact on the estimation of the cardiorespiratory interaction? Could it be necessary to study separated men and women?

·     3) Table 2 shows a statistical difference between the CRT groups for the BB for the follow-up condition. The CRT responders showed lower breathing frequency, and as the authors discussed in section 4.1, this could be because of a tissue hypoxia reduction. Then, it could be interesting to measure the SpO2 before and after the CRT procedure in both groups. What do you think about this suggestion?

·   4) It could be good to know how many of each group, HFSin, HFVES1, and HFVES2, are included in CRT responders and non-responders.

·      5) On page 4, line 160, an I (mutual information) is missed.

Author Response

The paper “Information-theoretic analysis of cardiorespiratory interactions in heart failure patients: effects of arrhythmias and cardiac re-synchronization therapy” describes the estimation of the cardiorespiratory coupling by linear and nonlinear approaches in patients suffering heart failure. The goals were to determine the effects of cardiac resynchronization therapy (CRT) on cardiorespiratory interaction and if this interaction could help to differentiate responders and non-responders to CRT in a preoperative phase. For that purpose, the ECG and the respiratory signal were measured immediately before CRT and after nine months. The paper is well-written and organized, representing a follow-up of a previous effort by the authors.  Some questions and comments are as follows:

According to reference 21, the CRT responders and non-responders were 62.9% and 80% men, respectively. Although there is no statistical difference in sex, according to Table 1 of the paper, the prevalence of men in the non-responders calls attention. Also, there is a statistical difference (p<0.02) for ischemic heart disease, with 50% in the non-responders group. What could the authors add about the influence of these two issues in the analyzed cardiorespiratory coupling?

The reviewer is  absolutely right. The patient sex and the etiology of heart failure are definitely one of the most important factors that determine the response to CRT and they are interrelated. In CRT landmark trials and in numerous subsequent studies, women comprise less than one-third of enrolled patients, but with better clinical outcomes of CRT in females, both in terms of mortality and heart failure endpoints. The question is whether the etiology of ischemic heart failure is a driver of sex difference in left ventricle remodeling after CRT implantation. Namely, women more often have non-ischemic cardiomyopathy. We know that in ischemic heart disease there is more fibrotic scar tissue, which localization and distribution is significantly less favorable, that impair conduction of electrical impulses generated by CRT more seriously, which minimizes the effects of the device on reverse remodeling of the left ventricle. Certainly, there are other factors that could suggest that female sex, regardless of etiology, contributes to better effects of CRT. First of all, women, unlike men, die more often from pump failure, so CRT will help them more. Then there are sex differences in electrical asynchrony. Females have more often true LBBB, and they have LBBB morphology at shorter QRS duration. Therefore, women display substrate for electrical resynchronization at shorter QRS duration, and since the same criteria for CRT implantation apply to both sexes, women with more serious conduction disorders will receive CRT more often. The smaller heart size in women is certainly an important factor in reverse remodeling after CRT implantation. Therefore, response to CRT is more pronounced in women but this therapy is still underused in this population.

Regarding acquiring the respiratory signal using a belt to measure abdominal movements, according to the literature, women have thoracoabdominal respiration, i.e., women and men have differences in respiratory movements. How does this impact on the estimation of the cardiorespiratory interaction? Could it be necessary to study separated men and women?

Definitely, there are important sex related differences in the structure and function (physiology) of the respiratory system. Women have smaller lungs and smaller large conducting airways than men and this has impact on resting pulmonary function, but also on response to exercise. Also, women have smaller ribcage size than men, with different inclination of the ribs (men’s ribs are more horizontally oriented than those of females), and shorter diaphragm. These are all reasons that explain greater contribution of inspiratory ribcage muscles in females than males during breathing (“in men ordinary breathing is chiefly by the diaphragm and in women chiefly by the ribs“). Also, sex hormones have great impact on the respiratory function. However, many authors have opinion that size, more than sex, is the main driving factor of the differences in respiratory function between men and women. Women are generally smaller than men, but in some studies when they examine women similar in size to men differences in respiratory movements and function disappeared. Therefore, we believe that sex is, for many reasons, an important factor when examining cardiopulmonary interaction, but that it is still only one of the factors, and that body mass index, age, fitness, current therapy, etc… probably also affect the interaction of these two systems. In other words, the potential differences in type of respiration in men and women are not sufficient reason to analyze cardiorespiratory interactions completely separately in relation to sex. We believe that the correct approach is to match study groups, whenever possible, by sex (and we have done it in previous research), but not to analyze men and women separately.

Table 2 shows a statistical difference between the CRT groups for the BB for the follow-up condition. The CRT responders showed lower breathing frequency, and as the authors discussed in section 4.1, this could be because of a tissue hypoxia reduction. Then, it could be interesting to measure the SpO2 before and after the CRT procedure in both groups. What do you think about this suggestion?

We believe that it is really a good idea. Nevertheless, we have certain concerns about it. Hypoxemia (low arterial oxygen saturation) does not always lead to tissue hypoxemia. Hypoxemia triggers compensatory mechanisms such as an increase in high concentration or a redistribution of cardiac output that enable the maintenance of adequate tissue oxygenation. Today we know that there are also compensatory mechanisms at cellular lever to allow cells to function in a hypoxic environment. Therefore, conclusions about changes in respiratory regulation, and especially cardiorespiratory interactions, that we would make based on changes in SpO2 would be questionable. In practice, blood pH or lactate values (signs of metabolic acidosis) are most often used to assess tissue hypoxia and perhaps we could use these analyzes in future research.

It could be good to know how many of each group, HFSin, HFVES1, and HFVES2, are included in CRT responders and non-responders.

This information is already provided in Table 1 and additional explanation is added in the revised version.

On page 4, line 160, an I (mutual information) is missed.

We thank the reviewer for noticing the typo, which has been corrected in the revised paper.

Reviewer 2 Report

Platisa et al. investigated CRC in heart failure patients applying different measures from information theory. Patients were divided in three groups depending on their heart rhythm (sinus rhythm and presence of low/high number of ventricular extrasystoles). They mainly discovered that the success of CRT is related to the number of extrasystoles and corresponding information transfer between the cardiac and respiratory signal quantified at baseline measurements, which could contribute to a better selection of patients for CRT therapy.

This is a very interesting manuscript for the readers of entropy, however, before acceptance some major revisions have to be performed.

The statistical procedures have to be standardized. All statistical analyses must be performed the non-parametric Mann-Whitney U, Wilcoxon pairwise test, and Friedman if necessary. And corrected for multiple testing if necessary.

The physiology of responders/non-responders to CRT therapy has to be explained in detail. Please show original data to support the understanding of the main results.

What is the effect of restoring RSA in CRT patients, what is the effect of VPC occurrence?

Author Response

Platisa et al. investigated CRC in heart failure patients applying different measures from information theory. Patients were divided in three groups depending on their heart rhythm (sinus rhythm and presence of low/high number of ventricular extrasystoles). They mainly discovered that the success of CRT is related to the number of extrasystoles and corresponding information transfer between the cardiac and respiratory signal quantified at baseline measurements, which could contribute to a better selection of patients for CRT therapy.

This is a very interesting manuscript for the readers of entropy, however, before acceptance some major revisions have to be performed.

The statistical procedures have to be standardized. All statistical analyses must be performed the non-parametric Mann-Whitney U, Wilcoxon pairwise test, and Friedman if necessary. And corrected for multiple testing if necessary.

We thank the reviewer for her/his suggestion. Nonparametric tests were performed in all comparisons. Corrected results are highlighted in blue in the revised manuscript.

The physiology of responders/non-responders to CRT therapy has to be explained in detail. Please show original data to support the understanding of the main results.

The profile of the patient who would be an ideal candidate for CRT, i.e. who we can say with certainty will benefit from this therapy, is still unknown. There are many factors that determine whether someone will be a CRT responder. Some of them are related to the patient’s characteristics, such as sex, age, etiology of heart failure, duration of the disease, number of previous hospitalizations due to heart failure, renal function, presence of arrhythmias, etc., and some depend on the position of coronary sinus lead (lead for stimulation of left ventricle), on programming the device and achieving high level of biventricular pacing (optimal function of the implanted device). In that context it is already known that premature ventricular ectopic beats burden above 0.1% on 24‐hour Holter, measured before CRT implantation, is associated with reduced overall biventricular pacing percentage and significantly worse outcome after device implantation. Therefore, the main result of this study is not that the success of CRT is related to the number of extrasystoles. It is a significant but not the main result, and we agree that it should not be highlighted, so we have now made changes to the conclusions and abstract in the revised paper. We believe that the main contribution of this paper is that we have shown that the success of CRT is related to corresponding information transfer between the cardiac and respiratory signal. After the repeated statistical analysis suggested by the reviewer, we obtained even stronger evidence for this conclusion and made changes to the results, especially the conclusions of the revised manuscript (lines 510-534). In the discussion we tried to give explanations for the obtained results, but we also pointed out that the knowledge so far is limited in certain areas, especially when it comes to the numerous compensatory mechanisms that are triggered in heart failure conditions, especially when extrasystoles are also present.

3) What is the effect of restoring RSA in CRT patients, what is the effect of VPC occurrence?

RSA is an index of cardiac vagal function. The restoring of RSA is directly related to the recovery of vagal tone and baroreceptor activity. The immediate effect of this is the decrease in heart rate. Then, restoring of RSA improves gas exchange because acceleration of heart rate during inspiration lead to increase in blood flow through the pulmonary circulation when alveoli are at peak oxygen concentration. This is valuable in patients with heart failure. RSA, by significantly contributing to heart rate variability (i.e. by increasing HRV), improves cardiac contractility, ventricular synchrony and coronary flow. In a broader context, the restoring of RSA indicates the restoration of balanced activity of control and compensatory mechanisms, which contributes to the stabilization of arterial blood pressure and systemic blood flow. From a clinical aspect, some studies have shown that a gradual deterioration of the ejection fraction can be seen in subjects with normal systolic function who have frequent ventricular extrasystoles. Although this topic requires a separate discussion, in the context of responding to CRT high premature ventricular contraction burden can disrupt the efficient function of a CRT system, reducing the frequency of effective biventricular pacing (PVC reduces percent of biventricular pacing, but also increases the presence of fusion or pseudofusion beats that can also be part of ineffective biventricular pacing) and as such, contributing to nonresponse. 

Reviewer 3 Report

This study investigated cardio-respiratory coupling (CRC) in patients with heart failure using information theory measures. The patients were categorized into groups based on the occurrence of ectopic beats, and their condition was examined both before and after undergoing cardiac resynchronization therapy (CRT).

The manuscript is well-structured, and the subject matter holds relevance in the context of selecting appropriate candidates for CRT. However, certain aspects require clarification:

 1. Abstract:

a) The abstract suggests a correlation between CRT success and the number of extrasystoles, as well as information transfer between cardiac and respiratory signals. However, the experimental design and results do not seem to fully support this claim, as the analysis of information theory measures with all three variables together (responders vs. non-responders, HF group, before vs. after) was not conducted.

2. Methods:

a) The first paragraph in section 2.1 refers to previous work by the same team by stating that “all patients were analyzed together”. However, the mentioned study already examined the differences between responders and non-responders in terms of cardiovascular coupling. The present study should explicitly outline the advancements it offers compared to previous research, both in the Methods section and the Discussion.

b) The classification of responders and non-responders was carried out in a previous study. Nonetheless, a brief description of how these groups were defined should be included here to ensure clarity.

c) The analysis does not specify which ECG lead or leads were selected. Is this selection relevant? Considering the ECG characteristics (such as noise and duration), would any lead enable accurate RR interval measurements.

d) The report does not mention the actual value of p, which represents the "memory" of the process. Does this value provide meaningful physiological insights? Would it be worthwhile to explore different p values?

3.    Discussion:

a) The initial paragraph in section 4.1 refers to the results in Table II. However, the statement that "resting heart and respiratory rates are higher in patients with ventricular arrhythmias and that the values of these parameters increase with the frequency of ventricular extrasystoles" does not appear substantiated, as no significant differences were observed between HF groups.

b) Line 387: What does EFLK stand for? Is it referring to ejection fraction (EF)

c) Line 478: "Therefore, it is obvious that the recovery of vagal tone and baroreceptor activity is responsible for strengthening the effect of respiration on cardiac rhythm, i.e., the intense cardiorespiratory interaction in this direction confirms the existence of balanced activity of control and compensatory mechanisms." It is not evident how this conclusion follows from the preceding sentences, which pertain to Fig. 4C and D. Perhaps this paragraph could be rephrased to enhance reader comprehension.

d) In the final paragraph of the discussion, CRT non-responders are identified as "patients showing a more pronounced ANS imbalance with a further increase in sympathetic activity." Could you clarify the reasoning behind this categorization? This question relates to my earlier comment in 2.b.

Typo: Footnote in Table I: "Respnders"

Author Response

This study investigated cardio-respiratory coupling (CRC) in patients with heart failure using information theory measures. The patients were categorized into groups based on the occurrence of ectopic beats, and their condition was examined both before and after undergoing cardiac resynchronization therapy (CRT).

The manuscript is well-structured, and the subject matter holds relevance in the context of selecting appropriate candidates for CRT. However, certain aspects require clarification:

  1. Abstract:
  2. a) The abstract suggests a correlation between CRT success and the number of extrasystoles, as well as information transfer between cardiac and respiratory signals. However, the experimental design and results do not seem to fully support this claim, as the analysis of information theory measures with all three variables together (responders vs. non-responders, HF group, before vs. after) was not conducted.

We thank the reviewer for his/her remarks. Abstract has been changed according to the provided suggestions.

  1. Methods:
  2. a) The first paragraph in section 2.1 refers to previous work by the same team by stating that “all patients were analyzed together”. However, the mentioned study already examined the differences between responders and non-responders in terms of cardiovascular coupling. The present study should explicitly outline the advancements it offers compared to previous research, both in the Methods section and the Discussion.

The reviewer is right: we outlined the advancements compared to previous research in the Methods section of the revised manuscript (page 3, lines 106-108).

  1. b) The classification of responders and non-responders was carried out in a previous study. Nonetheless, a brief description of how these groups were defined should be included here to ensure clarity.

We thank the reviewer for his/her remarks. The definition of the groups is added in the revised manuscript (page 3, lines 116-120).

  1. c) The analysis does not specify which ECG lead or leads were selected. Is this selection relevant? Considering the ECG characteristics (such as noise and duration), would any lead enable accurate RR interval measurements.

The reviewer is right, this selection is relevant but the Biopac equipment with MP100 unit can only measure one ECG channel and we measured ECG form lead 1. Data were acquired from patients monitored in the resting supine position and we had clear ECG signal without noise; therefore, RR interval measurement was not an issue in our protocol.

  1. d) The report does not mention the actual value of p, which represents the "memory" of the process. Does this value provide meaningful physiological insights? Would it be worthwhile to explore different p values?

We thank the reviewer for his/her comment. As reported in the original draft of the paper (lines 231-233), in the case of linear Granger causality estimation the order p of the regression models was set according to the Akaike Information Criterion, usually applied when dealing with the analysis of cardiorespiratory time series. As mentioned by the reviewer, the choice of the model order can influences the value of the achieved information measures and thus their physiological significance; this value should be sufficiently high to have enough 'memory' of the two processes involved in the interaction, in this case cardiac and respiratory activity, but, on the other hand, selecting an overly high model order leads to capturing fluctuations that are possibly not related to physiological mechanisms, e.g., to noise, thus leading to overfitting. For this reason, automatic statistically validated model order selection criteria like the one used in this study are employed to find the optimal compromise between the two tendencied mentioned (see e.g.: Faes, L., Erla, S., & Nollo, G. (2012). Measuring connectivity in linear multivariate processes: definitions, interpretation, and practical analysis. Computational and mathematical methods in medicine, 2012.). As regards the model free estimates of transfer entropy and cross entropy, also in this case the choice of the number of lagged components to be used to approximate the system past influences the measures of coupling and causality. In this case it must be emphasized that using large memories can lead to problems of bias in the estimation of the measures. For this reason, in this work we decided to set the embedding dimension to 2 (page 6, line 235), as commonly done in the literature when applying model free information-theoretic approaches to physiological time series (see e.g.: Faes L, Kugiumtzis D, Nollo G, Jurysta F, Marinazzo D. Estimating the decomposition of predictive information in multivariate 627 systems. Phys Rev E Stat Nonlin Soft Matter Phys. 2015;91(3):032904. Doi:10.1103/PhysRevE.91.032904).

  1. Discussion:
  2. a) The initial paragraph in section 4.1 refers to the results in Table II. However, the statement that “resting heart and respiratory rates are higher in patients with ventricular arrhythmias and that the values of these parameters increase with the frequency of ventricular extrasystoles” does not appear substantiated, as no significant differences were observed between HF groups.

We thank the reviewer for his/her remark. According to the suggestion of the first reviewer we applied nonparametric test analysis to all data comparisons. Then, we obtained significant difference between mean RR intervals in the HFSin and the HFVES2 group. This sentence has been changed.

  1. b) Line 387: What does EFLK stand for? Is it referring to ejection fraction (EF)

We thank the reviewer for his/her remark. Yes it is referring to ejection fraction of left ventricle. It is corrected in the revised manuscript.

  1. c) Line 478: “Therefore, it is obvious that the recovery of vagal tone and baroreceptor activity is responsible for strengthening the effect of respiration on cardiac rhythm, i.e., the intense cardiorespiratory interaction in this direction confirms the existence of balanced activity of control and compensatory mechanisms.” It is not evident how this conclusion follows from the preceding sentences, which pertain to Fig. 4C and D. Perhaps this paragraph could be rephrased to enhance reader comprehension.

According to the proper comment of the reviewer, the sentence was deleted in the revised paper.

  1. d) In the final paragraph of the discussion, CRT non-responders are identified as "patients showing a more pronounced ANS imbalance with a further increase in sympathetic activity." Could you clarify the reasoning behind this categorization? This question relates to my earlier comment in 2.b.

The reviewer is absolutely right that we do not have clear evidence for this claim in the results of this work (in our study, the fact that someone is a CRT non-responder does not mean that his heart failure is worsening, but that it is not improving to a sufficient degree). We could look for confirmation of these claim in the literature, but also in the results of our previous works (e.g. that in non-responders there is a significant reduction of short-term scaling exponent of RR interval series—α1(RR), a parameter that shows a strong negative correlation with the degree of sympathetic activity), but we fully agree that this should not be included in the discussion of this paper.

Typo: Footnote in Table I: "Respnders"

We thank the reviewer for noticing the typo, which has been corrected in the revised version of the manuscript.

Round 2

Reviewer 1 Report

The authors responded adequately to all my questions. I accept the paper in its present form.

Reviewer 3 Report

The authors have addressed all my previous concerns. I have no further comments.